# WHAT APPLES TELL ABOUT ORANGES: CONNECTING PRUNING MASKS AND HESSIAN EIGENSPACES

## ABSTRACT

Recent studies have demonstrated that good pruning masks of neural networks emerge *early* during training, and that they remain largely *stable* thereafter. In a separate line of work, it has also been demonstrated that the eigenspace of the loss Hessian shrinks drastically during *early* training, and remains largely *stable* thereafter. While previous research establishes a direct relationship between individual network parameters and loss curvature at training convergence, in this study we investigate the connection between parameter *pruning masks* and Hessian *eigenspaces*, throughout the entire training process and with particular attention to their *early stabilization*. To quantify the similarity between these seemingly disparate objects, we cast them as orthonormal matrices from the same Stiefel manifold, each defining a linear subspace. This allows us to measure the similarity of their spans using Grassmannian metrics. In our experiments, we train an *unpruned* deep neural network and demonstrate that these two subspaces overlap significantly—well above random chance—throughout the entire training process and not just at convergence. This overlap is largest at initialization, and then drops and stabilizes, providing a novel perspective on the early stabilization phenomenon and suggesting that, in deep learning, largest parameter magnitudes tend to coincide with the directions of largest loss curvature. This early-stabilization and high-overlap phenomenon can be leveraged to approximate the typically intractable top Hessian subspace via parameter inspection, at only linear cost. The connection between parameters and loss curvatures also offers a fresh perspective on existing work, tending a bridge between first- and second-order methods.

## 1 INTRODUCTION

Deep neural networks (DNNs) often benefit from a large number of parameters; but not all parameters are equally important. Frequently, a substantial portion of the weights can be *pruned*, i.e. removed, during or at the end of training, without compromising the model's performance (see Blalock et al., 2020; Hoefler et al., 2021). One efficient method to identify these pruned subnetworks is via the parameter's magnitude (Han et al., 2015). Interestingly, these subnetworks materialize very early in training (Frankle & Carbin, 2019), and once they emerge, their topology stops changing significantly (Achille et al., 2019; You et al., 2020). In other words, competitive subnetworks *crystallize early* in training and remain *stable* thereafter (Section 2.2).

A parallel line of research focuses on the Hessian matrix of the loss, which characterizes the loss landscape's curvature. Among other things, the Hessian is used to understand generalization (e.g. Hochreiter & Schmidhuber, 1997; Keskar et al., 2017), improve neural network training (e.g. Martens, 2016; Rodomanov & Nesterov, 2021), tune hyperparameters (e.g. LeCun et al., 1992; Cohen et al., 2021), tackle overconfidence (e.g. Kristiadi et al., 2021), or prune networks (e.g. LeCun et al., 1989). Multiple studies determined empirically that the Hessian spectrum separates into two parts: the *bulk* subspace, with near-zero eigenvalues, and the *top* subspace with significantly larger eigenvalue magnitudes (e.g. Dauphin et al., 2014; Sagun et al., 2018). Importantly, Gur-Ari et al. (2018) reported that, after a few initial training steps, the gradient predominantly lies within the top subspace. Moreover, the top eigenspace remains relatively stable throughout training. Analogously to pruning, this line of research indicates that the top Hessian eigenspace *crystallizes early* in training and tends to remain *stable* throughout the training (Section 2.1).

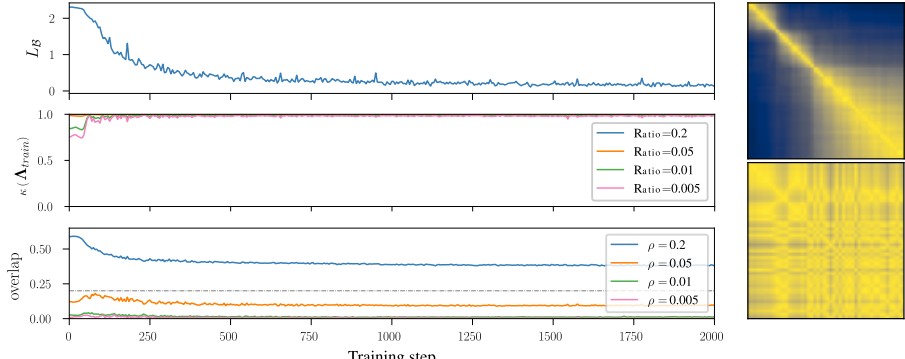

Figure 1: **Pruning masks and top Hessian subspaces exhibit early crystallization and stabilization. Additionally, both spaces show a similarity well above random chance.** *(left)* The minibatch loss $L_{\mathcal{B}}$, followed by the early compression of the Hessian spectrum $\mathbf{\Lambda}$ and the overlap between parameter magnitude pruning masks and top Hessian subspaces (bottom). Note how overlap is well above random chance (illustrated by the dotted gray line for $\rho = 0.2$), and how it stabilizes after an initial decay. *(right)* The depicted distance matrices showcase the stabilization of both parameter pruning masks in terms of their Intersection over Union (IoU) (top) and top Hessian subspaces in terms of their overlap (bottom), as reported in (You et al., 2020) and (Gur-Ari et al., 2018). See Section 5 and Figures 4, 9 and 10 for extended results and interpretations.

In this work, we explore the similarity between these two—currently largely independent—lines of research, noticing that both report early emergence and subsequent stabilization of much smaller sub-structures. Such a study is necessary because, while the Optimal Brain Damage (OBD) framework (LeCun et al., 1989) establishes a direct link between individual parameter magnitudes and loss curvature, it does so at training convergence, and hence does not address early emergence or stabilization of pruning masks and Hessian subspaces. On the other hand, train-time analysis of the Hessian typically focuses on global or layer-wise parameter groups (e.g. Papyan, 2020; Sankar et al., 2021), and it does not connect loss curvatures to arbitrary parameter subsets. We propose a method to connect parameter *pruning masks* to *Hessian eigenspaces* in a training-sensitive and stabilization-aware manner. Specifically, we make the following contributions:

1. We characterize pruning masks $\boldsymbol{m}_k$ (which select $k$ parameters and discard the rest), as rank-$k$ orthogonal matrices (Section 3), belonging to the same Stiefel manifold as any matrix formed by $k$ Hessian eigenvectors. This allows us to directly compare both their spans via *Grassmannian metrics*.

2. We review popular Grassmannian metrics and analyze their computational and statistical properties. We empirically identify the overlap metric as advantageous (Section 4).

3. We provide empirical evidence that the similarity between spaces induced by $\boldsymbol{m}_k$ magnitude pruning masks (obtained from a network that has *not* been pruned) and top-$k$ Hessian eigenspaces is substantially larger than random chance throughout the whole training process. This overlap is largest at initialization, then drops and stabilizes, providing a novel perspective on the early stabilization phenomenon and suggesting that, in deep learning, largest parameter magnitudes tend to coincide with the directions of largest loss curvature (Figure 1 and Section 5).

There are multiple ways in which a connection between Hessian eigenspaces and parameter masks may be useful: (1) High similarity between parameter magnitudes and loss curvatures may be an indicator of common underlying phenomena, supporting the research of an *early phase of neural network training* (Frankle et al., 2020b) and complementing the Fisher Information Matrix approach used by Achille et al. (2019). (2) Since pruning masks can be obtained in linear time, our results suggest new ways for fast and effective low-rank Hessian approximations, with application to e.g. pruning and optimization methods as proposed by Hassibi et al. (1993); Gur-Ari et al. (2018). Early stabilization can also be further leveraged, as discussed in, e.g. You et al. (2020). (3) If parameters carry information about loss curvatures, novel characterizations of existing methods may be possible e.g. by bridging the gap between first- and second-order methods.

## 2 BACKGROUND AND RELATED WORK

We consider a supervised classification or regression setup with a training set $\mathcal{D}_{\text{train}} :=$ $\{(\boldsymbol{x}_n, \boldsymbol{y}_n)\}_{n=1}^N$ of labeled data $(\boldsymbol{x}_n, \boldsymbol{y}_n) \in \mathbb{X} \times \mathbb{Y}$, stemming from an unknown true data-generating distribution $P$. The DNN $f_{\boldsymbol{\theta}}(\boldsymbol{x}) : \mathbb{X} \to \mathbb{Y}$ maps inputs $\boldsymbol{x}$ to predictions $\hat{\boldsymbol{y}}$ via parameters $\boldsymbol{\theta} \in \mathbb{R}^D$. A loss function $\ell : \mathbb{Y} \times \mathbb{Y} \to \mathbb{R}_{\geqslant 0}$ penalizes differences between prediction $\hat{\boldsymbol{y}}$ and true label $\boldsymbol{y}$. The goal is then to minimize the inaccessible *risk* $L_P(\boldsymbol{\theta}) := \int \ell(f_{\boldsymbol{\theta}}(\boldsymbol{x}), \boldsymbol{y}) dP$ via the proxy *empirical risk* $L_{\mathcal{D}_{\text{train}}}(\boldsymbol{\theta}) := \frac{1}{N} \sum_{n=1}^N \left(\ell(f_{\boldsymbol{\theta}}(\boldsymbol{x}_n), \boldsymbol{y}_n)\right)$. For typically large $N$, we can approximate $L_{\mathcal{D}_{\text{train}}}(\boldsymbol{\theta})$ using *mini-batches* $\mathcal{B} \overset{\text{iid}}{\approx} \mathcal{D}_{\text{train}}$ of $B \ll N$ samples. Furthermore, we assume that $f$ and $\ell$ are twice differentiable, which allows us to access the *gradient vector* $\boldsymbol{g}(\boldsymbol{\theta}) := \nabla_{\boldsymbol{\theta}} L(\boldsymbol{\theta}) \in \mathbb{R}^D$, indicating the direction of steepest increase for the local loss, and the *Hessian matrix* $\boldsymbol{H}(\boldsymbol{\theta}) := \nabla_{\boldsymbol{\theta}}^2 L(\boldsymbol{\theta}) \in \mathbb{R}^{D \times D}$, providing information about the curvature of the local loss landscape[1].

### 2.1 THE HESSIAN IN DEEP LEARNING

As outlined in Section 1, $\boldsymbol{H}$ plays a prominent role across a wide range of deep learning (DL) applications, showcasing its significance. A useful characterization of $\boldsymbol{H}$ is through its eigendecomposition $\boldsymbol{H} := \boldsymbol{U} \boldsymbol{\Lambda} \boldsymbol{U}^\top = \sum_{i=1}^D \lambda_i \boldsymbol{u}_i \boldsymbol{u}_i^\top$. Here, $\boldsymbol{U}$ is orthogonal, with column *eigenvectors* $\boldsymbol{u}_i$, and $\boldsymbol{\Lambda}$ is diagonal and real-valued with *eigenvalues* $|\lambda_1| \geqslant \ldots \geqslant |\lambda_D|$. We call $\boldsymbol{U}^{(k)} := \{\boldsymbol{u}_i\}_{i=1}^k$ the *top-k eigenbasis* of $\boldsymbol{H}$, and $\text{span}(\boldsymbol{U}^{(k)})$ the *top-k eigenspace*. A relevant result is that the top-$k$ eigendecomposition $\boldsymbol{H}^{(k)} := \sum_{i=1}^k \lambda_i \boldsymbol{u}_i \boldsymbol{u}_i^\top$ minimizes $\|\boldsymbol{H} - \boldsymbol{H}^{(k)}\|$ for all unitarily invariant norms (Golub & Van Loan, 2013).

Recent literature has extensively investigated the Hessian **spectrum** of DNNs, revealing that the eigenvalues are typically clustered into at least two parts: (1) the *bulk* of eigenvalues with near-zero magnitude and (2) a few *top* eigenvalues that have significantly larger magnitude (e.g. Sagun et al., 2018; Papyan, 2019). These top eigenvalues display interesting traits, such as *class/cross-class* covariance structures in classification tasks (Papyan, 2020), or training regimes with $\lambda_1$ often hovering around $\frac{2}{\eta}$, in the so-called *edge of stability* (Cohen et al., 2021). As for the Hessian **eigenspace**, Li et al. (2018) showed that projecting the whole space onto a few random, fixed dimensions still allows Stochastic Gradient Descent (SGD) to perform competitively—provided enough dimensions are given—leading to the idea of an *intrinsic dimensionality* of problems. In contrast, (Gur-Ari et al., 2018) observed that *this restriction to a lower-dimensional, fixed subspace seems to happen spontaneously anyway*: after a brief initial period of training, the gradient predominately lies within a small subspace spanned by the few top Hessian eigenvectors and this space changes little over the remaining training process. Still, some authors note that some of the observed phenomena are heavily reliant on specific optimizer and model choices Li et al. (2018); Ghorbani et al. (2019).

One fundamental issue that greatly limits the scope and scale of DL experiments that can be done is that $\boldsymbol{H}$ is extremely large—typically prohibitive—which renders the computation of many related quantities unfeasible. As a consequence, most scalable methods are *matrix-free*, and rely on Hessian-Vector Products (HVPs) to compute linear maps of $\boldsymbol{H}$ in linear memory and time (Pearlmutter, 1994). Examples are the computation of individual rows/columns, traces (Hutchinson, 1989), diagonal entries (Becker & LeCun, 1989; Martens et al., 2012), spectral densities (Yao et al., 2020; Papyan, 2018), and top-$k$ Hessian eigenpairs (Golub & Van Loan, 2013; Halko et al., 2011). To make those more accessible, specialized DL libraries have been developed recently to provide broader access to these Hessian quantities (e.g. Dangel et al., 2020; Yao et al., 2020; Elsayed & Mahmood, 2022), but this is an active field of research, and efficiently accessing further Hessian quantities remains a major challenge. An example of this, relevant for our work, is that in order to link strong directions of curvature with arbitrary parameters, one needs to access arbitrary sub-matrices of $\boldsymbol{U}$ (since rows of $\boldsymbol{U}$ are associated with specific parameters, and columns with specific eigenvalues).

### 2.2 NEURAL NETWORK PRUNING

Pruning involves removing parameters while maintaining performance. By eliminating irrelevant weights, the pruned model is smaller, more computationally efficient, and may even converge faster

---

[1]In general, with $\boldsymbol{H}$ we refer to *any* Hessian of the loss. If we want to emphasize the data domain, we use a subindex, e.g. $\boldsymbol{H}_{\mathcal{B}}$ refers to the Hessian of the mini-batch loss $L_{\mathcal{B}}$.

and generalize better (e.g. Gale et al., 2019; Blalock et al., 2020; Hoefler et al., 2021). In DL, pruning is typically characterized via element-wise multiplication of parameter vector $\boldsymbol{\theta}$ with a boolean *pruning mask* $\boldsymbol{m}_k \in \mathbb{B}^D$ with $k$ entries with a value of 1, and 0 for the rest. This yields a pruned neural network $f_{\boldsymbol{m}_k \odot \boldsymbol{\theta}}(\boldsymbol{x})$, where a subset of the parameters is permanently fixed to 0. We refer to a mask as *$k$-sparse* when it has exactly $k$ non-zero elements, and define the ratio $\rho(\boldsymbol{m}) := \Sigma_i\, m_i / D$ as a measure of sparsity. For non-boolean vectors, we rather want to measure whether a small subset of parameters $\boldsymbol{\theta}_\iota$ contains a large proportion of the energy. When the subset $\iota$ is assumed to be known, this can be directly expressed as the ratio: $\kappa(\boldsymbol{\theta})^2 := \|\boldsymbol{\theta}_\iota\|_2^2 / \|\boldsymbol{\theta}\|_2^2$ (see Hurley & Rickard (2009) for other cases and further discussion).

Competitive performances can be often achieved with $\rho \sim [1\%, 10\%]$ (Hoefler et al., 2021), but this is dependent on the task, model and pruning method. Various methods have been proposed to determine good pruning masks, which differ mainly in the *ranking criterion*, i.e. determining weights to prune, and the *pruning schedule*, i.e. determining when to prune. *Structured* pruning methods use an architecture-specific ranking criterion; *unstructured* methods are architecture-agnostic (e.g. Liu et al., 2017). One simple, competitive and widely used unstructured ranking criterion is *magnitude pruning*, which involves removing parameters with the smallest absolute values (Han et al., 2015).

A pivotal observation was that good pruning masks can already exist upon initialization (Frankle & Carbin, 2019) or very early in training (Frankle et al., 2020a). Training only these subnetworks from scratch can yield competitive performance. This phenomenon is known as the *Lottery Ticket Hypothesis (LTH)*. Nonetheless, *identifying* such "winning tickets" remains a hard task. Recent work has further demonstrated that parameter magnitude rankings and thus pruning masks stabilize early in training. You et al. (2020) compared Hamming distances between periodically extracted pruning masks and found that they stop changing early in training, yielding the *Early-Bird Lottery Tickets (EBLTs)*. This phenomenon is in line with the *loss of information plasticity* reported in Achille et al. (2019), and resonates with the theme of *early emergence followed by stabilization* that was also observed for the top Hessian subspace and that is central to our work.

## 3 PRUNING AS ORTHOGONAL PROJECTIONS

Our focus is to quantify the connection between pruning masks and top-$k$ Hessian eigenspaces, since both are reported to undergo early crystallization and stabilization. We thus require a way to relate a mask to a subspace. In this section, we show that both $\boldsymbol{U}^{(k)}$ and a $k$-sparse[2] mask $\boldsymbol{m}_k$ can be characterized as elements of the same compact Stiefel manifold $\mathbb{O}^{D \times k} := \{\boldsymbol{Q} : \boldsymbol{Q} \in \mathbb{R}^{D \times k},\ \boldsymbol{Q}^\top \boldsymbol{Q} = \boldsymbol{I}_k\}$, where $\boldsymbol{I}_k$ is the rank-$k$ identity matrix (Absil et al., 2004). We can then ask: *how similar are the spans of such elements?*

**Reordering parameters:** Recall from Section 2.1 that we assumed Hessian eigenvalues sorted by descending magnitude, exposing a single cutting point between *top* and *bulk* eigenspace at dimension $k$. To simplify notation, we impose a similar condition, by defining a permutation matrix $\boldsymbol{P}$ for any given mask $\boldsymbol{m}_k$ such that the mask entries are grouped in *selected* (i.e. all $k$ entries where $\boldsymbol{m}$ equals 1) and *discarded* (i.e. all indexes where $\boldsymbol{m}$ equals 0), offering also a single cutting point: $\tilde{\boldsymbol{m}} := \boldsymbol{P}^\top \boldsymbol{m} = (1, 1, 1, \ldots 0, 0, 0)$. Without loss of generality, we permute the parameters $\tilde{\boldsymbol{\theta}} = \boldsymbol{P}^\top \boldsymbol{\theta}$ accordingly, as well as the rows and columns of the Hessian $\tilde{\boldsymbol{H}} := \boldsymbol{P}^\top \boldsymbol{H} \boldsymbol{P} = \tilde{\boldsymbol{U}} \boldsymbol{\Lambda} \tilde{\boldsymbol{U}}^\top$. Thus, $(\boldsymbol{m}, \boldsymbol{\theta}, \boldsymbol{H}) \cong (\tilde{\boldsymbol{m}}, \tilde{\boldsymbol{\theta}}, \tilde{\boldsymbol{H}})$ is an isomorphism, $\boldsymbol{H}$ and $\tilde{\boldsymbol{H}}$ are *similar*, and the loss curvature remains unaltered ($\tilde{\boldsymbol{\theta}}^\top \tilde{\boldsymbol{H}} \tilde{\boldsymbol{\theta}} = \boldsymbol{\theta}^\top \boldsymbol{H} \boldsymbol{\theta}$).

**Masking as an orthogonal projection:** Using $\boldsymbol{P}$, any $k$-sparse masking can be expressed as:

$$\boldsymbol{P}^\top (\boldsymbol{m}_k \odot \boldsymbol{\theta}) = \tilde{\boldsymbol{m}}_k \odot \tilde{\boldsymbol{\theta}} =: \tilde{\boldsymbol{\Phi}}_k \tilde{\boldsymbol{\theta}} := \begin{pmatrix} \boldsymbol{I}_k & 0 \\ 0 & 0 \end{pmatrix} \tilde{\boldsymbol{\theta}} = \begin{pmatrix} \boldsymbol{I}_k \\ 0 \end{pmatrix} \begin{pmatrix} \boldsymbol{I}_k & 0 \end{pmatrix} \tilde{\boldsymbol{\theta}} =: \boldsymbol{I}_{D,k} \boldsymbol{I}_{D,k}^\top \tilde{\boldsymbol{\theta}} \qquad (1)$$

Note that $\boldsymbol{I}_{D,k} \in \mathbb{B}^{D \times k}$, which we define as the subset of $\mathbb{O}^{D \times k}$ with boolean entries.

**Partitioning $\tilde{\boldsymbol{H}}$:** Consider now the following partition of the reordered Hessian:

---

[2]This can be extended to masks with other sparsities using Schubert varieties (Ye & Lim, 2016).

$$\tilde{H} := \underbrace{\begin{pmatrix} V & \vdots & W \\ \hdashline \bar{V} & \vdots & \bar{W} \end{pmatrix}}_{P^\top \tilde{U}} \underbrace{\begin{pmatrix} D & \vdots & \\ \hdashline & \vdots & \bar{E} \end{pmatrix}}_{\Lambda} \underbrace{\begin{pmatrix} V^\top & \vdots & \bar{V}^\top \\ \hdashline W^\top & \vdots & \bar{W}^\top \end{pmatrix}}_{\tilde{U}^\top P}, \qquad \begin{matrix} V, D \in \mathbb{R}^{k \times k} \\ \bar{W}, \bar{E} \in \mathbb{R}^{(D-k) \times (D-k)} \end{matrix} \qquad (2)$$

With this partition, $\tilde{U}^{(k)} =: \begin{pmatrix} V \\ \bar{V} \end{pmatrix}$ is the *top-k* Hessian eigenbasis, whereas $\begin{pmatrix} W \\ \bar{W} \end{pmatrix}$ represents the *bulk* eigenbasis. Conversely, the rows of $(V|W)$ correspond to the *selected* parameters, and $(\bar{V}|\bar{W})$ to the *discarded* ones. Since $\tilde{U}$ is orthogonal, so is $\tilde{U}^{(k)}$, and thus an element of $\mathbb{O}^{D \times k}$. Note how this partition exposes the interaction between top eigenspace and arbitrary parameter subsets.

## 4 MEASURING SIMILARITY OF SUBSPACES VIA GRASSMANNIAN METRICS

Given that both $k$-sparse pruning masks and the top-$k$ Hessian eigenspace can be cast as elements of the same Stiefel manifold, we now want to quantify their similarity. Specifically, we are only interested in the similarity of their spanned *spaces*, not the particular basis, as two distinct elements of $\mathbb{O}^{D \times k}$ can have the same span, e.g. by permuting their columns. This problem is addressed by Grassmannian metrics, which we review in Section 4.1, and analyze in more depth in Section 4.2, finding that overlap provides an informative, stable, popular and potentially scalable metric.

### 4.1 GRASSMANN MANIFOLDS AND THEIR METRICS

Grassmann manifolds are extensively studied (Bendokat et al., 2020) and find relevant application in various fields like physics (e.g. Witten, 1988), numerics (e.g. Absil et al., 2004) and, more recently, DL (e.g. Gur-Ari et al., 2018; Zhang et al., 2018; Dangel et al., 2022). A Grassmann manifold $\mathcal{G}_{k,D}$ is the set of all $k$-dimensional subspaces of a given $D$-dimensional Euclidean space. Two matrices $(Q_i, Q_j) \in (\mathbb{O}^{D \times k} \times \mathbb{O}^{D \times k})$ map to the same element $\mathfrak{g} \in \mathcal{G}_{k,D}$ if and only if they have the same span. Therefore, *the subset of all matrices in $\mathbb{O}^{D \times k}$ that map to $\mathfrak{g}_i := \mathrm{span}(Q_i)$ forms an equivalence class* $\mathcal{S}_i^{\mathbb{O}} := \{Q_j : Q_j Z_j = Q_i, Z_j^\top Z_j = I_k\}$ (e.g. Edelman et al., 1998; Absil et al., 2004).

A desirable property of Grassmann manifolds is the availability of closed-form expressions for their *geodesics* (i.e. the shortest paths between any two elements in $\mathcal{G}$) and metric functions. We can thus measure distances between subspaces in an interpretable and computationally amenable manner: geodesics from $\mathfrak{g}_i$ to $\mathfrak{g}_j$ follow circular trajectories, and therefore their *distance* can be interpreted as the "amount of rotation" needed to go from one space to another. Such rotations can be succinctly expressed in terms of *principal angles* $\sigma_{i \to j} \in [0, \frac{\pi}{2}]^k$, and they can be efficiently obtained from $\mathbb{O}^{D \times k}$ matrices via their Singular Value Decomposition (SVD):

$$Q_i^\top Q_j =: L_{i \to j} \operatorname{diag}\left( \cos(\sigma_{i \to j}) \right) R_{i \to j}^\top, \quad L, R \text{ orthogonal.} \qquad (3)$$

Since $Q_i, Q_j$ have orthonormal columns, $\cos(\sigma_{i \to j}) \in [0,1]^k$ (e.g. Neretin, 2001), and more similar spans will yield larger singular values, which translate to smaller rotations. Importantly, singular values are invariant under similarity:

$$\operatorname{diag}\left( \cos(\sigma_{i \to j}) \right) = L_{i \to j}^\top Q_i^\top Q_j R_{i \to j} = L_{i \to j}'^\top (Z_i^\top Q_i^\top)(Q_j Z_j) R_{i \to j}'. \qquad (4)$$

In other words, they are invariant under the action of $Z$, which means that $\sigma$ does not change if we replace an input matrix with any other matrix from the same equivalence class (see definition for $\mathcal{S}_i^{\mathbb{O}}$). The family of functions that satisfy this invariance, plus the axioms of metric spaces, form the family of *Grassmannian metrics* (e.g. Qiu et al., 2005), each capturing a different notion of distance between subspaces (e.g. largest principal angle vs. sum of principal angles). In the following, we highlight popular metrics from the literature (e.g. Edelman et al., 1998). We abbreviate $\operatorname{dist}_*(\mathfrak{g}_i, \mathfrak{g}_j) = f(\sigma_{i \to j})$ as $\operatorname{dist}_* = f(\sigma)$, where $*$ refers to any unitarily invariant norm:

a) **Geodesic distance:** This is the *arc length* of the geodesic between the respective spaces in $\mathcal{G}$, defined as $\operatorname{dist}_g = \|\sigma\|_2 \in [0, \frac{\pi}{2}\sqrt{k}]$.

b) **Chordal norm:** $\operatorname{dist}_{c,*}$, obtained by minimizing $\|Q_i Z_i - Q_j Z_j\|_*$ over orthogonal matrices $(Z_1, Z_2)$ (for that reason it is also called *Hausdorff distance*). The $\ell_2$ and $\ell_F$ norms admit a closed-form solution in terms of principal angles: $\operatorname{dist}_{c,2} = \|2\sin\left(\frac{1}{2}\sigma\right)\|_\infty \in [0, \sqrt{2}]$ and $\operatorname{dist}_{c,F} = \|2\sin\left(\frac{1}{2}\sigma\right)\|_2 \in [0, \sqrt{2k}]$.

**c) Projection norm:** Also called the *gap metric*, it uses the unique orthogonal projector representation of a given subspace, i.e. $\mathbf{\Psi}_i = \mathbf{Q}_i\mathbf{Q}_i^\top$, as follows: $\mathrm{dist}_{p,*} = \|\mathbf{\Psi}_i - \mathbf{\Psi}_j\|_*$. Here, we also have closed-form expressions for the $\ell_2$ and $\ell_F$ norms: $\mathrm{dist}_{p,2} = \|\sin(\boldsymbol{\sigma})\|_\infty \in [0,1]$ and $\mathrm{dist}_{p,\mathrm{F}} = \|\sin(\boldsymbol{\sigma})\|_2 \in [0, \sqrt{k}]$.

**d) Fubini-Study:** This quantity is a measure of the *acute angle* between both spaces, generalized to higher dimensions: $\mathrm{dist}_a = \arccos\left(|\det(\mathbf{Q}_i^\top\mathbf{Q}_j)|\right) = \arccos\left(\prod_i \cos(\sigma_i)\right) \in [0, \frac{\pi}{2}]$.

**e) Overlap:** The overlap $= \frac{1}{k}\|\mathbf{\Psi}_i\mathbf{Q}_j\|_F^2 \in [0,1]$ quantity was used in Gur-Ari et al. (2018) to measure subspace *similarity*. It is not a metric *per se*, since it is highest for equivalent subspaces and decreases with their distance, but it is a bijection of $\mathrm{dist}_{p,\mathrm{F}}$, as follows: $\frac{1}{k}\|\mathbf{\Psi}_i\mathbf{Q}_j\|_F^2 = \frac{1}{k}\|\mathbf{Q}_i^\top\mathbf{Q}_j\|_F^2 = \frac{1}{k}\|\cos(\boldsymbol{\sigma})\|_F^2 = 1 - \|\cos(\boldsymbol{\sigma})\|_F^2 = 1 - \frac{1}{k}\mathrm{dist}_{p,F}^2$.

While the above metrics apply to any pair of matrices from $\mathbb{O}^{D\times k}$, there are also relevant metrics specific to $\mathbb{B}^{D\times k}$, that can be characterized in a similar manner. Consider an arbitrary pair of $k$-sparse matrices $(\boldsymbol{m}_i, \boldsymbol{m}_j)$, and their corresponding orthogonal projectors $\mathbf{\Phi} := \mathrm{diag}(\boldsymbol{m})$. Then we have:

**i) IoU:** Typically used as an evaluation metric, it is defined as the relative number of entries present in *both* masks, i.e. $\mathrm{IoU} := \frac{\boldsymbol{m}_i \cap \boldsymbol{m}_j}{\boldsymbol{m}_i \cup \boldsymbol{m}_j} \in [0,1]$. Then we have $\boldsymbol{m}_i \cap \boldsymbol{m}_j = \|\mathbf{\Phi}_i\mathbf{\Phi}_j\|_F^2 = k \cdot \mathrm{overlap}$, and if both masks are $k$-sparse, we also have $\boldsymbol{m}_i \cup \boldsymbol{m}_j = 2k - (\boldsymbol{m}_i \cap \boldsymbol{m}_j) = k(2 - overlap)$, yielding the bijection $\mathrm{overlap} = \frac{2\,\mathrm{IoU}}{1+\mathrm{IoU}}$.

**ii) Hamming distance:** This quantity, defined as the minimum number of bit-flips needed to pass from one mask to another, was used in You et al. (2020) to measure distances between pruning masks. It is in fact a Grassmannian metric: $\mathrm{dist}_\mathrm{H} := \|\boldsymbol{m}_i - \boldsymbol{m}_j\|_2^2 = \|\mathbf{\Phi}_i - \mathbf{\Phi}_j\|_F^2 = \mathrm{dist}_{p,\mathrm{F}}^2 \in [0, k]$, which means that the bijection $\mathrm{overlap} = 1 - \frac{1}{k}\mathrm{dist}_\mathrm{H}$ also holds.

## 4.2 Comparing Grassmannian metrics

We now conduct a synthetic experiment to identify the most insightful Grassmannian metric for comparing pruning masks and Hessian eigenspaces. The experiment also provides baselines to discern when a given metric value can be considered "larger than chance". We also explore computational costs associated with each metric and discuss their potential scalability for larger-scale DL scenarios.

**Setup:** We compute the reviewed Grassmannian metrics **a)** to **e)** between randomly drawn matrices from $\mathbb{O}^{k\times D}$ and masks from $\mathbb{B}^{k\times D}$. We use the subgroup algorithm (Diaconis & Shahshahani, 1987) to sample matrices uniformly from $\mathbb{O}^{k\times D}$ and column permutations of $\boldsymbol{I}_{D,k}$ to sample uniformly from $\mathbb{B}^{k\times D}$. To determine if the metrics behave differently for masks than for general orthogonal matrices, we inspect three different *modalities*: pairs of matrices ($\mathbb{O}$-to-$\mathbb{O}$), masks ($\mathbb{B}$-to-$\mathbb{B}$), and matrix-mask pairs ($\mathbb{O}$-to-$\mathbb{B}$). We normalize all metrics, denoted by $\mathrm{dist}_{\overline{*}}$, to be in $[0,1]$, with 1 indicating highest similarity (i.e. smallest distance). We want to inspect how the distribution of the metrics changes as a function of height $D$ and width-to-height ratio $r := \frac{k}{D}$. First, given fixed values of $r$, we study the distribution of all Grassmannian metrics as a function of $D$ (Figure 2). Additionally, given fixed values of $D$ we study how the Grassmannian metrics change as a function of $r$ (Figure 3). Appendix A.2 provides supplementary details, including figures for the other modalities.

**All distributions become *predictable* as $D$ increases:** The expectation of all metrics *converges to a fixed value as $D$ increases* (Figure 2). Furthermore, the variance of all observed distributions tends to shrink. While results are unstable for very low $D$, they stabilize already for moderate values of $D$, well below the typical DL regime. Thus, *we can use the measured "converged" expectations as random baselines to compare against* (see Table 1 in Appendix A.2). Furthermore, it can be shown analytically that the expectation of overlap equals $\frac{k}{D}$ (a proof is included in Appendix A.1).

**Metrics display either *shrinking* or *proportional* behavior as $D$ increases:** Although all distributions become predictable, not all of them are informative. The top row in Fig. 2 showcases the *shrinking* metrics ($\mathrm{dist}_{\overline{c,2}}$, $\mathrm{dist}_{\overline{p,2}}$ and $\mathrm{dist}_{\overline{a}}$) whose expectation collapses to zero as $D$ increases. On the contrary, all metrics portrayed in the bottom row exhibit *proportional* behavior, with their expectation converging to a value that seems to only depend on $r$. Mathematically, all shrinking metrics are dominated by the *extremal* principal angles, whereas proportional metrics rely on a balanced aggregation of all angles.

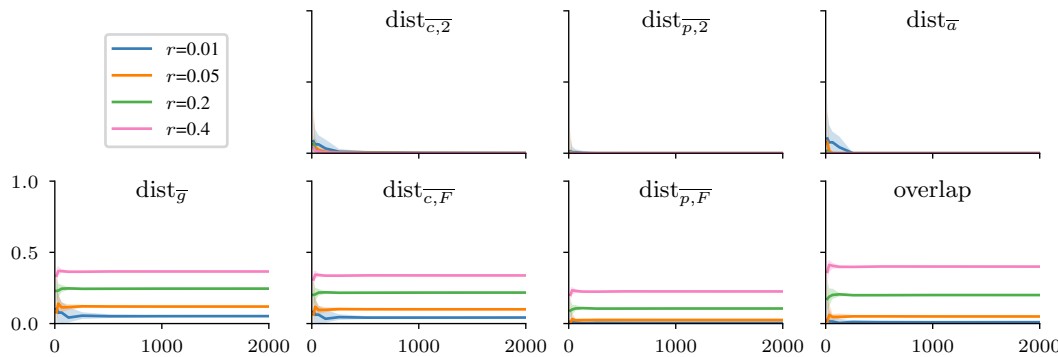

Figure 2: **Grassmannian metrics for random pairs of matrices** $(\boldsymbol{Q}_1, \boldsymbol{Q}_2) \overset{\text{unif.}}{\sim} (\mathbb{O}^{D \times k}, \mathbb{O}^{D \times k})$ **as a function of** $D$**.** Each subplot shows a different metric, and each color corresponds to a different ratio $r = \frac{k}{D}$. For each $(D, k)$, we sample 50 random pairs and report the median of the resulting distribution (line plot) as well as the 5-95 percentiles (shaded regions). See Section 4.2 for details.

**Mask-vs-mask metrics have larger variance, and in some cases lower expectations:** All distributions for the mask-vs-mask modality have higher variance compared to the other modalities (see Appendix A.2). Furthermore, the distributions for the $\text{dist}_{\overline{g}}$ and $\text{dist}_{\overline{c,F}}$ metrics seem to follow lower trajectories for the mask-vs-mask modality, compared to the other modalities. This is not the case for $\text{dist}_{\overline{p,F}}$ and overlap, whose expectation does not seem to be affected by the modality.

**Extremal values of $r$ lead to saturation, except for overlap:** Most metrics exhibit a nonlinear behavior near the extremes (Figure 3). This is particularly extreme for shrinking metrics, but saturation can also be observed in the proportional metrics. The only exception is overlap, whose expectation is linear as shown in Appendix A.1.

**Computational aspects:** Given $\boldsymbol{Q}_i$ and $\boldsymbol{Q}_j$, all metrics reviewed in Section 4.1 can be expressed in terms of *principal angles*, which can be computed at the cost of a *thin matrix multiplication* and an *SVD* as presented in Eq. (3). The Hausdorff distances can also be alternatively formulated as an optimization objective, but this is not in closed form and the runtime depends on the optimization procedure. The matrix formulation of projection norms is in closed form, but it involves large $D \times D$ projector matrices. The $\mathbb{B}^{D \times k}$ metrics can be expressed in terms of element-wise mask operations, which can be computed in linear time and memory. The real bottleneck in our computations is obtaining one of the involved matrices, namely $\tilde{\boldsymbol{U}}^{(k)}$, since the number of parameters even in realistic DL setups is prohibitive even for $k$ in the dozens.

**A case for overlap:** As a result of the above discussion, we consider overlap the most suitable metric for our purposes, since: *(1) It is informative:* Given a fixed sparsity (as is the case in our work), its uniformly random behavior is predictable (i.e. converging and low-variance) as well as informative (i.e. non-shrinking). Furthermore, its expectation behaves linearly, and we know analytically that it equals $\frac{k}{D}$. *(2) It is computationally efficient:* While it still requires to obtain $\tilde{\boldsymbol{U}}^{(k)}$, its matrix formulation involves thin matrices only, allowing us to avoid the use of large $D \times D$ projector matrices and the SVD computation, which is one reason it has been used in related literature (e.g. Gur-Ari et al., 2018; Dangel et al., 2022). *(3) It is related to other metrics:* overlap can be mapped to other popular metrics such as the Hamming distance, IoU, or $\text{dist}_{p,F}$ via bijections (see Section 4.1).

## 5    PRUNING MASKS AND HESSIAN EIGENSPACES OVERLAP SIGNIFICANTLY

We now investigate the similarity between parameter magnitude pruning masks and top Hessian eigenspaces, as a function of training progress. Figure 1 highlights the main results of this section, and comprehensive results are reported in Figs. 4, 9 and 10.

**Setup:** As discussed in Section 4.2, all reviewed metrics require knowledge of the Hessian eigenspace. Due to the computational costs involved in obtaining $\tilde{\boldsymbol{U}}^{(k)}$, we have to limit the size of the neural network that we use. At the same time, we need an network that is sufficiently large to allow for the existence of redundant parameters. A sweet spot is provided by the Multi-Layer Percep-

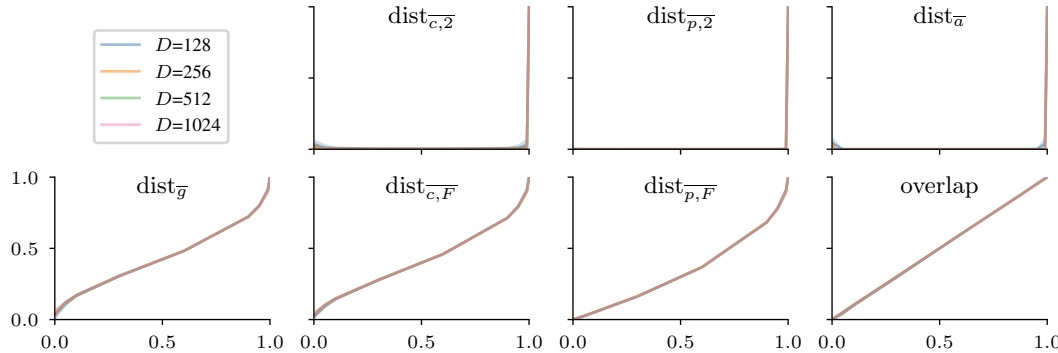

Figure 3: **Grassmannian metrics for random pairs of matrices** $(Q_1, Q_2) \overset{\text{unif.}}{\sim} (\mathbb{O}^{D \times k}, \mathbb{O}^{D \times k})$ **as a function of** $r = \frac{k}{D}$**.** Each subplot shows a different metric, and each color corresponds to a different dimension $D$. For each $(D, k)$, we sample 50 random pairs and report the median of the resulting distribution (line plot) as well as the 5-95 percentiles (shaded regions). See Section 4.2 for details.

tron (MLP) from Martens & Grosse (2015) which, featuring only 7030 parameters, is able to achieve zero training loss when trained on a $16 \times 16$ subsampled version of MNIST. We borrow the same setup, but use SGD with a learning rate of 0.3 to train the model, achieving a test accuracy of 96.4% after 50 epochs. At initialization, we isolate 500 samples from the training set and 500 from the validation set to compute the respective Hessians $H_{train}$ and $H_{test}$. During training, for each step $t \in \{0, 5, 10, \ldots, 2000\}$, we record the network parameters $\theta^{(t)}$, as well as $H_{train}^{(t)}, H_{test}^{(t)}$. Then, we define $m_k^{(t)}$ to be 1 for the $k$-largest parameters by magnitude, and 0 otherwise. At each step $t$, we compute the Grassmannian metrics between the span of $m_k^{(t)}$ and the top eigenspace $\tilde{U}_{train}^{(k)}{}^{(t)}$. For any two steps $t_i, t_j$, we also compute $\text{IoU}(m_k^{(t_i)}, m_k^{(t_j)})$, $\text{overlap}(\tilde{U}_{train}^{(k)}{}^{(t_i)}, \tilde{U}_{train}^{(k)}{}^{(t_j)})$ and $\text{overlap}(\tilde{U}_{test}^{(k)}{}^{(t_i)}, \tilde{U}_{test}^{(k)}{}^{(t_j)})$ . Note that we use the pruning masks to define the subspace, but we do not apply them, i.e. *we do not prune but simply observe the potential pruning masks*. This is desirable: if we pruned the model during training, we would be restricting the parameter space, which could artificially boost the overlap between $m_k$ and $\tilde{U}^{(k)}$.

**Both subspaces experience early collapse and stabilization:** As we can see in Figure 9, the sparsity metric $\kappa$ for both $\theta$ and $\Lambda$ raises, and a minority of the support quickly ends up condensing a majority of the norm. Furthermore, we observe that the pairwise similarities between subsequent magnitude pruning masks, as well as top Hessian eigenspaces, remain high after only a few training steps (Figure 10 and left and center subplot of Figure 1). Thus, our setup experiences the scenario of *early emergence followed by stabilization* highlighted in Section 2.

**Both subspaces overlap significantly:** Interestingly, computing the Grassmannian metrics between the pruning masks and the top Hessian subspaces reveals that there is a significant similarity between these two spaces (Figure 4 and right subplot of Figure 1). The overlap between them is much larger than random chance, i.e. the random baseline established in the previous section. We further observe that the overlap is largest directly at initialization, and then decays but stabilizes well above random chance. This is in resonance with the idea of loss of plasticity (Achille et al., 2019).

## 6 CONCLUSION

We started with the observation that, at the early stages of neural network training, both pruning masks and the loss Hessian eigenspace undergo drastic simplification and subsequent stabilization. To investigate a possible connection between these two phenomena, we proposed a principled methodology to quantify their similarity, in the form of Grassmannian metrics, and isolated overlap as a particular advantageous one. Experiments using this metric support the notion of an *early phase* in training. They also reveal a striking similarity between magnitude pruning masks and top Hessian eigenspaces well above chance level, suggesting that in DL large parameters tend to coincide with directions of high loss curvature. This connection opens the door to the development of novel optimization and pruning methods, and provides a new angle for the analysis of existing ones.

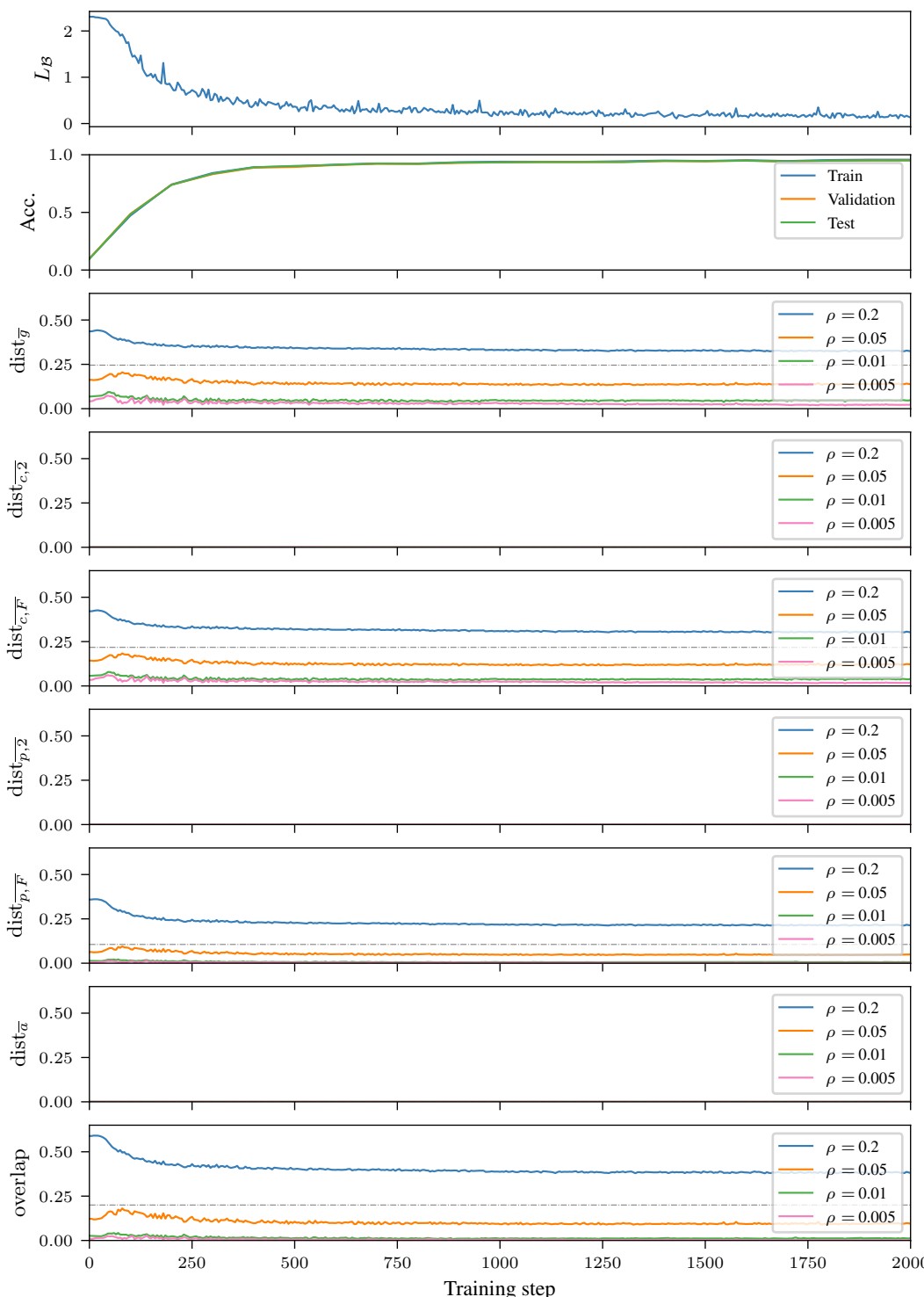

Figure 4: **Different Grassmannian metrics between pruning masks and top Hessian subspaces.** Each line shows a particular sparsity level $\rho$, i.e.. the ratio of unpruned parameters or the size of the top Hessian subspace relative to the full Hessian. All *proportional* metrics reveal a significant similarity between spaces spanned by pruning masks and top Hessian subspaces well above random chance (random baselines gathered from our synthetic experiments are shown in gray dotted lines for $\rho = 0.2$), while *shrinking* metrics are effectively zero .

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

## A    SUPPLEMENTARY MATERIAL

### A.1    EXPECTATION OF overlap FOR UNIFORMLY RANDOM MATRICES

In Section 4.2 we empirically observed that the expectation under uniformly distributed random matrices becomes *predictable* for all reviewed metrics. Here we show that, for overlap, such expectation equals exactly $\frac{k}{D}$. This lemma depends on a standard calculation that was communicated to us by Joel A. Tropp.

**Lemma A.1.** *Let $Q_1, Q_2$ be random matrices drawn uniformly from the Stiefel manifold $\mathbb{O}^{D \times k} := \{Q : Q \in \mathbb{R}^{D \times k}, Q^\top Q = I_k\}$. Then,*

$$\mathbb{E}\left[\text{overlap}(\text{span}(Q_1), \text{span}(Q_2))\right] = \frac{k}{D} \tag{5}$$

*Proof.* We start by rewriting the definition of overlap, presented in Section 4.1, in terms of the trace of orthogonal projectors $\Psi = QQ^\top$. We make use of the idempotence of $\Psi$ and the unitary invariance of the Frobenius norm:

$$\text{overlap}(\text{span}(Q_1), \text{span}(Q_2)) = \frac{1}{k}\|Q_1^\top Q_2\|_F^2 = \frac{1}{k}\|\Psi_1 \Psi_2\|_F^2 = \frac{1}{k}\text{Tr}(\Psi_1 \Psi_2^2 \Psi_1) \tag{6}$$

$$= \frac{1}{k}\text{Tr}(\Psi_1 \Psi_2 \Psi_1) \tag{7}$$

We further observe that, if $Q$ is drawn uniformly, the marginal distribution of every column $q$ is the uniform distribution over the Euclidean unit sphere, hence it is isotropic:

$$\mathbb{E}\left[qq^\top\right] = \frac{1}{D}I_D \tag{8}$$

Where $I_D$ is the rank-$D$ identity matrix. Then, leveraging linearity of expectation, the orthogonal projector $\Psi = QQ^\top$ can be expressed as follows:

$$\mathbb{E}\left[\Psi\right] = \sum_{i=1}^{k}\mathbb{E}\left[q_i q_i^\top\right] = \frac{k}{D}I_D \tag{9}$$

Now, given two independent realizations $Q_1, Q_2$, we form the associated orthogonal projectors $\Psi_1, \Psi_2$. Write $\mathbb{E}_1, \mathbb{E}_2$ for the expectations of the respective distributions of $Q_1$ and $Q_2$. Then, leveraging independence of $Q_1$ and $Q_2$, idempotence of $\Psi$, and linearity of expectation and trace, we have:

$$\mathbb{E}\left[\text{Tr}(\Psi_1 \Psi_2 \Psi_1)\right] = \mathbb{E}_1\left[\text{Tr}(\Psi_1 \mathbb{E}_2\left[\Psi_2\right]\Psi_1)\right] = \frac{k}{D}\mathbb{E}_1\left[\text{Tr}(\Psi_1 I_D \Psi_1)\right] = \frac{k^2}{D^2}\text{Tr}\,I_D = \frac{k^2}{D} \tag{10}$$

Replacing in the original definition concludes the proof.    $\square$

### A.2    SYNTHETIC EXPERIMENT ON GRASSMANNIAN METRICS

This section provides details about the synthetic experiment presented in Section 4.2. Algorithm 1 details the overall procedure. We used the following values:

- Number of random (matrix or mask) samples: $T = 50$
- For Figures 2, 5 and 6 we investigate four different (fixed) ratios $r \in \{0.4, 0.2, 0.05, 0.01\}$ at several increasing dimensions $d \in \{16, 32, 64, 128, 256, 512, 1024, 2048\}$.
- For Figures 3, 7 and 8 we investigated four (fixed) dimensions $d \in \{128, 256, 512, 1024\}$ at several increasing ratios $r \in \{0.005, 0.01, 0.05, 0.1, 0.33, 0.66, 0.9, 0.95, 0.99, 1\}$.

Note how, in the second case, we concentrate the width-to-height ratios around the extremes. This is to better capture the behavior of *shrinking* metrics, as discussed in Section 4.2.

### A.3    ADDITIONAL RESULTS FOR SECTION 5

---

**Algorithm 1:** Synthetic experiment on Grassmannian metrics (see Section 4.2 for details).

**Input:** $\{D_1, D_2, ...\}$                                                        // Matrix height ($D_i \in \mathbb{N}$)
**Input:** $\{r_1, r_2, ...\}$                                        // Width-to-height ratio ($r_i \in [0,1]$)
**Input:** $\{(\mathbb{O}, \mathbb{O}), (\mathbb{O}, \mathbb{B}), (\mathbb{B}, \mathbb{B})\}$                                                              // Modality
**Input:** $\{\text{dist}_{\overline{g}}, \text{dist}_{\overline{c,2}}, \text{dist}_{\overline{c,F}}, \text{dist}_{\overline{p,2}}, \text{dist}_{\overline{p,F}}, \text{dist}_{\overline{a}}, \text{overlap}\}$   // Metric (normalized)
**Input:** $T$                                            // Number of random samples

1  $\mathcal{R} \leftarrow \varnothing$                                            // Result (a dictionary)
2  **for** $d \in \{D_1, D_2, ...\}$ **do**
3    **for** $r \in \{r_1, r_2, ...\}$ **do**
4      $k \leftarrow \max(1, \text{round}(r \cdot d))$
5      **for** $\text{dist} \in \{\text{dist}_{\overline{g}}, \text{dist}_{\overline{c,2}}, \text{dist}_{\overline{c,F}}, \text{dist}_{\overline{p,2}}, \text{dist}_{\overline{p,F}}, \text{dist}_{\overline{a}}, \text{overlap}\}$ **do**
6        **for** $(\mathcal{M}_1, \mathcal{M}_2) \in \{(\mathbb{O}, \mathbb{O}), (\mathbb{O}, \mathbb{B}), (\mathbb{B}, \mathbb{B})\}$ **do**
7          $\mathcal{H} \leftarrow \varnothing$                                // Collection of samples
8          **for** $\{1, ..., T\}$ **do**
9            $(\boldsymbol{Q}_1, \boldsymbol{Q}_2) \overset{\text{unif.}}{\sim} (\mathcal{M}_1^{d \times k}, \mathcal{M}_2^{d \times k})$ $\mathcal{H} \leftarrow \mathcal{H} \cup \text{dist}\left(\text{span}(\boldsymbol{Q}_1), \text{span}(\boldsymbol{Q}_2)\right)$
10         **end**
11         $\mathcal{R}_{[d, r, \text{dist}, \mathcal{M}_1, \mathcal{M}_2]} \leftarrow \mathcal{H}$                       // Gather samples into result
12       **end**
13     **end**
14   **end**
15 **end**
16 **return** $\mathcal{R}$

---

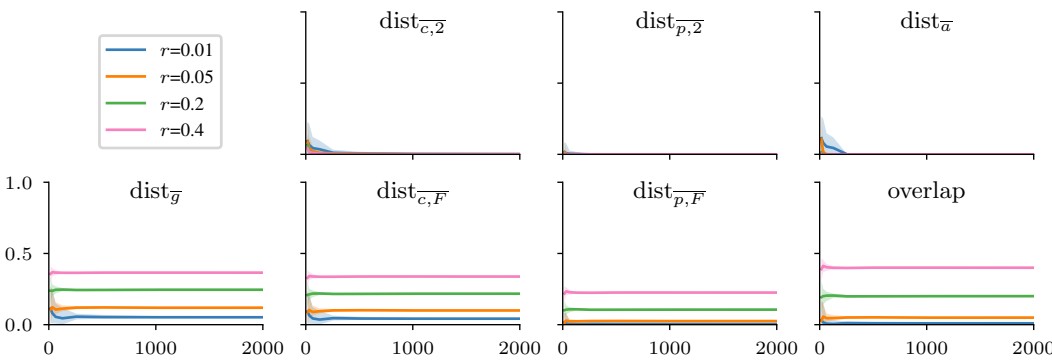

Figure 5: **Grassmannian metrics for random pairs of matrices and masks** $(\boldsymbol{Q}_1, \boldsymbol{Q}_2) \overset{\text{unif.}}{\sim}$ $(\mathbb{O}^{D \times k}, \mathbb{B}^{D \times k})$ **as a function of** $D$ **and for fixed** $r$**.** Each subplot shows a different Grassmannian metric with the different lines indicating four different ratios $r$. For each value of $D$, we report the median metric over 50 random pairs with the shaded regions showing the 5-95 percentiles.

Table 1: **Random baselines for Grassmannian metrics.** Shown are the measured expectations (averaged over 50 samples at $D = 2048$) of different Grassmannian metrics between two uniformly random matrices in $\mathbb{O}$. Ostensibly, they only depend on the width-to-height ratio $r$ (see Section 4.2).

| Metric | $r$ | | | | |
|---|---|---|---|---|---|
|  | 0.005 | 0.01 | 0.05 | 0.2 | 0.4 |
| $\text{dist}_{\overline{g}}$ | 0.96289 | 0.94809 | 0.88091 | 0.75466 | 0.63464 |
| $\text{dist}_{\overline{c,2}}$ | 0.99803 | 0.99839 | 0.99952 | 0.99954 | 0.99974 |
| $\text{dist}_{\overline{c,F}}$ | 0.97017 | 0.95798 | 0.90016 | 0.78246 | 0.66217 |
| $\text{dist}_{\overline{p,2}}$ | 0.99999 | 0.99999 | 1.0 | 1.0 | 1.0 |
| $\text{dist}_{\overline{p,F}}$ | 0.99752 | 0.99521 | 0.97487 | 0.89430 | 0.77473 |
| $\text{dist}_{\overline{a}}$ | 1.0 | 1.0 | 1.0 | 1.0 | 1.0 |
| overlap | 0.00495 | 0.00955 | 0.04962 | 0.20022 | 0.39980 |

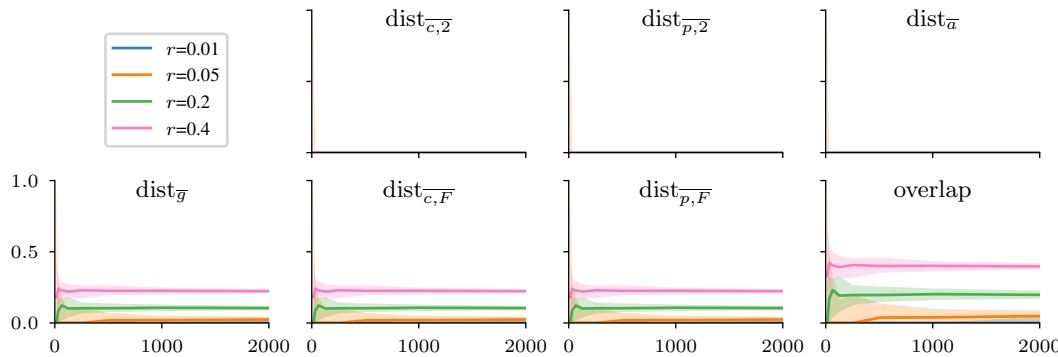

Figure 6: **Grassmannian metrics for random pairs of masks** $(Q_1, Q_2) \overset{\text{unif.}}{\sim} (\mathbb{B}^{D \times k}, \mathbb{B}^{D \times k})$ **as a function of $D$ and for fixed $r$.** Each subplot shows a different Grassmannian metric with the different lines indicating four different ratios $r$. For each value of $D$, we report the median metric over 50 random pairs with the shaded regions showing the 5-95 percentiles.

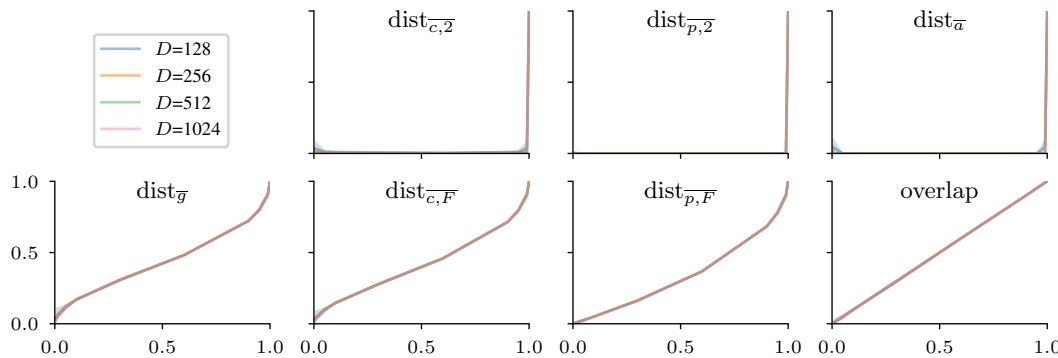

Figure 7: **Grassmannian metrics for random pairs of matrices and masks** $(Q_1, Q_2) \overset{\text{unif.}}{\sim} (\mathbb{O}^{D \times k}, \mathbb{B}^{D \times k})$ **as a function of $r$ and for fixed $D$.** Each subplot shows a different Grassmannian metric with the different lines indicating four different dimensions $D$. For each value of $r$, we report the median metric over 50 random pairs with the shaded regions showing the 5-95 percentiles.

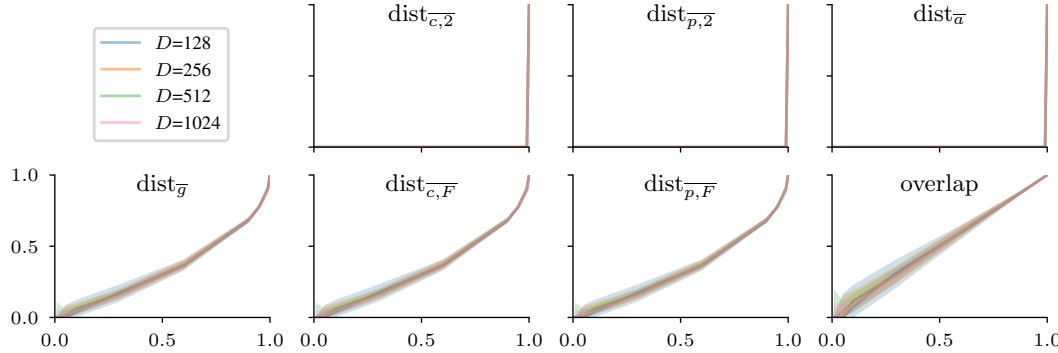

Figure 8: **Grassmannian metrics for random pairs of masks** $(Q_1, Q_2) \overset{\text{unif.}}{\sim} (\mathbb{B}^{D \times k}, \mathbb{B}^{D \times k})$ **as a function of $r$ and for fixed $D$.** Each subplot shows a different Grassmannian metric with the different lines indicating four different dimensions $D$. For each value of $r$, we report the median metric over 50 random pairs with the shaded regions showing the 5-95 percentiles.

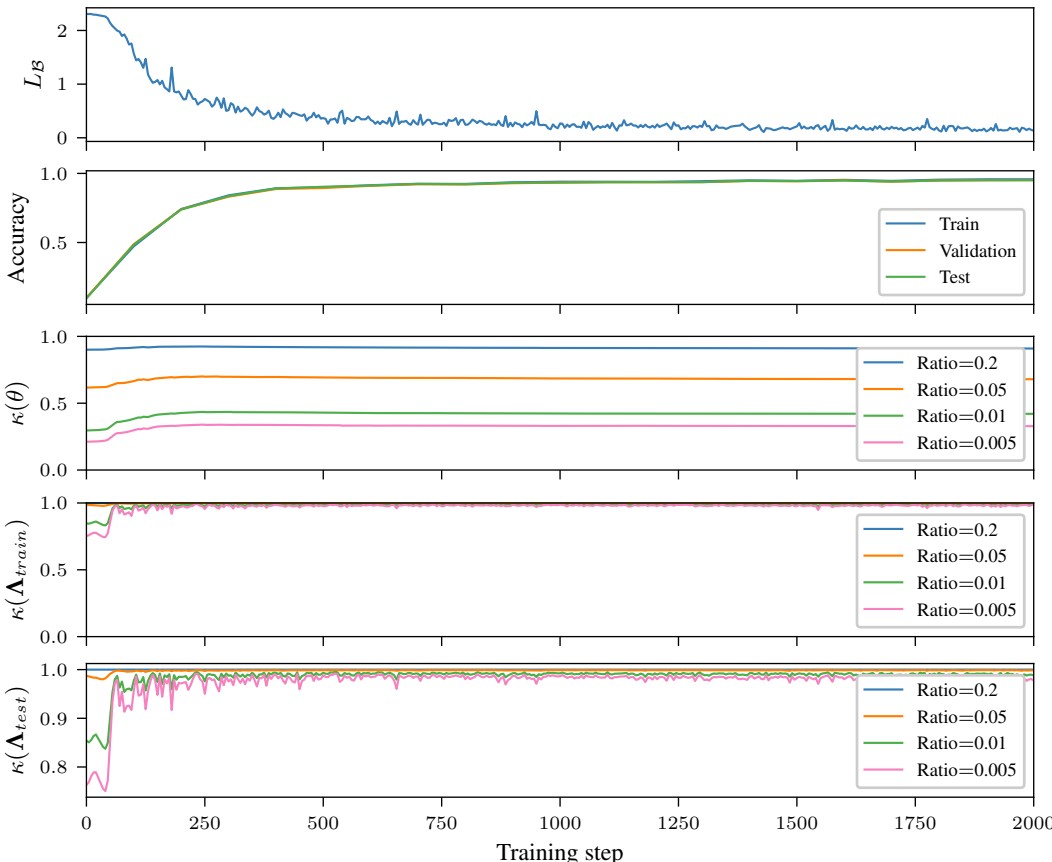

Figure 9: **Both the parameters of the a neural network as well as their Hessian spectrum collapse early during training.** *(top)* The top two subplots show the mini-batch loss $L_\mathcal{B}$ as well as the train/validation/test accuracy of the model trained in Section 5. *(middle)* Looking at the top $20\%, 5\%, 1\%, 0.5\%$ of parameters by magnitude, we can see that very early during training, most of the energy is concentrated on a small subset of the parameters (see Section 2 for a definition of $\kappa$). For example, shortly after initialization, the top $0.5\%$ largest parameters by magnitude have roughly $1/4$ of the total magnitude of all parameters. *(bottom)* We can observe a similar behavior for the Hessian spectrum on both the training set (fourth subplot) and the test set (fifth subplot). Only a few steps after training, most of the energy is concentrated in only $0.5\%$ of the eigenvalues.

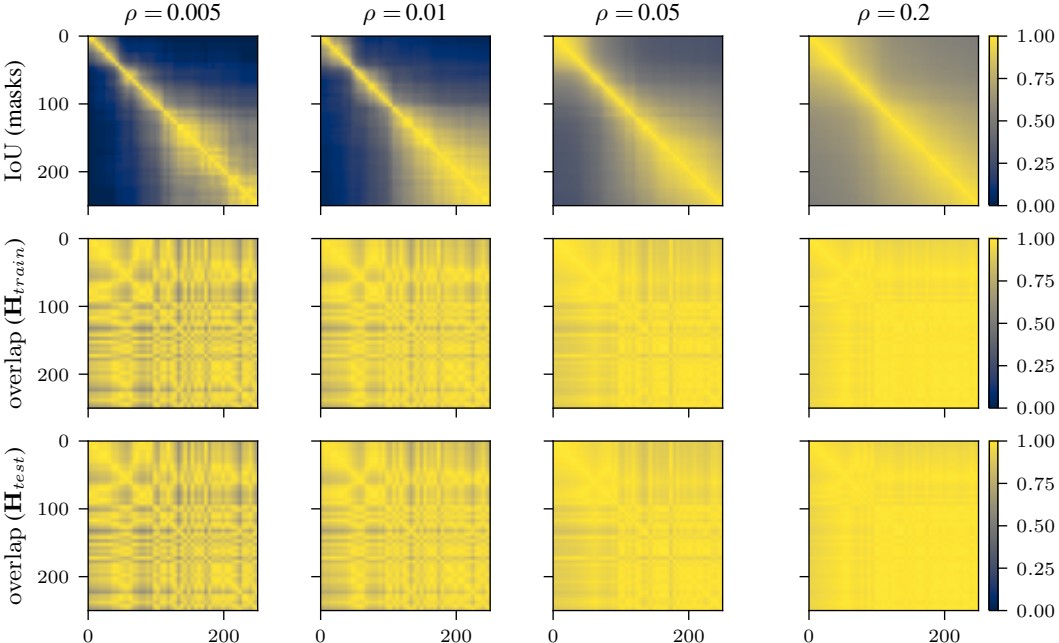

Figure 10: **Both parameter pruning masks (top row) and Hessians (bottom two rows) remain relatively stable after an initial training phase.** Following You et al. (2020), the depicted matrices represent all pairwise similarities (i.e. higher is better) from the beginning of training (top left corners) until step 240 (bottom right corners). For this reason, all matrices are symmetric and have unit diagonals. *(top)* Even when selecting only $0.5\%$ of the parameters (left column), masks collected at different training steps show a remarkable similarity after an initial phase of training. *(middle and bottom)* The overlap metric for top Hessian eigenspaces on the train (middle) and test set (bottom) extracted at different training steps and for different subspace sizes (columns). Starting from initialization, the Hessian eigenspaces do not change significantly over the course of training. No substantial differences in behaviour between $\boldsymbol{H}_{test}$ and $\boldsymbol{H}_{train}$ can be observed.

