# OpenReview forum: "What Apples Tell About Oranges: Connecting Pruning Masks and Hessian Eigenspaces"
_ICLR.cc/2024/Conference — Submitted to ICLR 2024_

### Official Review · Reviewer_e5Zh · 2023-10-31

**Soundness:** 2 fair
**Presentation:** 3 good
**Contribution:** 2 fair
**Rating:** 3
**Confidence:** 3

**Summary:**

The authors aim to draw parallels between the early stabilization of pruning masks during training and the shrinking of the Hessian eigenspace early in training. In order to do so, they cast the sparse mask with $k$ nonzero parameters and the top-k Hessian eigenvectors as matrices in the same Stiefel manifold. This operation allows them to compare the spans of the two matrices using an overlap Grassmannian metric. With the proposed method, the authors observe an above random overlap between the pruning mask and the hessian eigenspace, which is large initially and stabilizes during training. The authors conclude that such a similarity suggests that large weight magnitudes correspond to large curvatures of the loss landscape.

**Strengths:**

The proposed method of using a Grassmannian metric to compare the pruned mask and the Hessian eigenspaces by casting them in a Stiefel manifold is an interesting and seemingly useful proposal.

**Weaknesses:**

However, I have several concerns regarding the claims and the conclusions that the author’s draw in the paper.

1. The authors do not take into account the effect of different pruning criteria on the overlap similarity. I believe the pruning criteria itself might have an important role to play in determining the amount of overlap between the hessian eigenspace and the pruning masks. For example, are there above random overlaps for random pruning or using a iterative pruning using the SNIP criterion [1]. Such an experiment would also shed more light on the significance of the overlap.

2. The authors only show empirical results on the MNIST datasets which I think is insufficient. In order to confirm the presence of such an overlap, the authors must consider multiple datasets. If the computation of the eigenvectors of the Hessian is computationally problematic for larger image datasets, they can consider smaller tabular/algorithmic datasets.

3. At high sparsities, the overlap is not significant (Fig. 4 has lower overlap than the random baseline for pruning ratio < 0.2). The authors have not addressed this sufficiently in the paper.

4. The authors claim that the overlap between the masks and hessian eigenspaces suggests that large weights correspond to large curvatures in the loss. But this claim is not verified. For example, for a pruning criteria that retains the smallest weights instead of the largest, does the overlap still hold. Is the overlap affected by a SAM [2] like regularizer? Such ablations would be essential to verify the connection between weight magnitude and loss landscapes. Moreover, for homogenous activations, parameters can be arbitrarily scaled without changing the function but only modifying the loss curvature. This can also potentially change the behaviour of magnitude pruning and hence the overlap.


[1] Lee, Namhoon, Thalaiyasingam Ajanthan, and Philip Torr. "SNIP: SINGLE-SHOT NETWORK PRUNING BASED ON CONNECTION SENSITIVITY." International Conference on Learning Representations. 2018.

[2] Foret, Pierre, et al. "Sharpness-aware Minimization for Efficiently Improving Generalization." International Conference on Learning Representations. 2020.

**Questions:**

The authors have established an interesting methodology to compare and study pruning masks in context of Hessian eigenspaces. However, in its current state, the author’s have not sufficiently verified this connection in the paper.

---

### Official Review · Reviewer_FPBX · 2023-10-31

**Soundness:** 3 good
**Presentation:** 3 good
**Contribution:** 2 fair
**Rating:** 3
**Confidence:** 3

**Summary:**

This paper investigates the connection between the Hessian spectrum of the loss and the masks associated with magnitude pruning. Several similarity metrics are introduced and discussed, and a simple experiment shows that indeed there is a small albeit significant overlap between the subspaces spanned by the pruning masks and the top eigenvectors of the Hessian.

**Strengths:**

This paper raises an intriguing hypothesis on the connection between pruning masks unveiled via magnitude pruning and the spectrum of the Hessian. The hypothesis is original, relevant, and well-motivated. The methodology is introduced clearly and explored thoroughly, so that  it might serve as a valuable reference for future research on the topic.

**Weaknesses:**

The main shortcoming of the paper lies in the scarcity of conclusive results, which are obtained from the proposed method.

First, I find it hard to understand the relevance of some of the properties that the authors used to evaluate their metrics. For instance, what are the implications of the dependence of the measure's variance on the modality? What are the implications of the expectation of overlap not depending on the modality?

Secondly, the observed correlation between the Hessian spectrum and pruning outcomes seems to decrease with training, then saturate at a relatively small value. Can the authors suggest a reason behind this phenomenon?

Finally, the experimental evaluation is limited to a single case study, raising questions about the practical significance of the relationship between Hessian and Pruning. The paper could greatly benefit from a more comprehensive analysis of the findings and a discussion of potential future directions.

**Questions:**

1. What is v (argument of rho) in the first paragraph of section 2.2?

2. What is the value of k in figure 4? Is the overlap providing more insight than say dist_{p,F} or dist_{c,F}?

---

### Official Review · Reviewer_ZWvF · 2023-10-31

**Soundness:** 3 good
**Presentation:** 3 good
**Contribution:** 2 fair
**Rating:** 5
**Confidence:** 3

**Summary:**

In this work, the authors conduct an empirical investigation into the connection between pruning masks and the eigenspace of the Hessian of the loss function of a neural network. To do so, they show that pruning masks with a given number of nonzero elements $K$ and rank-$K$ approximations of the Hessian lie on the same Stiefel manifold. The authors then empirically investigate a variety of Grassmanian metrics, and use those metrics that they identify to be useful, to show that the Hessian eigenspace and the pruning mask are similar (by those metrics, such as $\mathrm{overlap}(\cdot)$).

**Strengths:**

The paper has several strengths:

* This work proposes an interesting idea, that the Hessian eigenspace and pruning masks are similar.
* Moreover, they show that the similarity is "stable" early in training/
* The observation that the pruning masks and the low-rank approximations of the Hessian lie on the same Stiefel manifold is a simple but elegant way to illustrate the connection between the two seemingly disparate quantities.
* A variety of metrics on the Grassmanian manifold are investigated, and useful metrics are clearly identified.
* The paper itself is nicely written.

**Weaknesses:**

While the paper has several strengths, it has a few key weaknesses as well.

* Given the fact that this paper is an empirical investigation, the investigation into the paper's key claim - that Hessian eigenspaces and pruning masks are similar - is perhaps a little insubstantial. The claim appears to have been investigated only for a very small network on MNIST. In the absence of rigorous theoretical results, a deeper investigation on different models and tasks would significantly strengthen the case made in the paper. While the computational constraints are quite clear, perhaps using approximate Hessians, or layerwise or even filter-wise Hessians would have helped (see, for instance, [1],[3]).
* Second-order methods have been used for pruning in prior work ([1],[2],[4], [5]). A thorough investigation into how the observations made in this paper reconcile with prior work would have been of significant interest to the community.

[1] *WoodFisher: Efficient second-order approximations for model compression*. Singh and Alistarh, 2020.

[2] *Group Fisher Pruning for Practical Network Compression*. Liu et al, 2021.

[3] *Analytic Insights into Structure and Rank of Neural Network Hessian Maps*. Singh et al, 2021

[4] *Optimal Brain Surgeon and general network pruning*. Hassibi et al, 1994

[5] *Optimal Brain Damage*. le Cun, 1989.

**Questions:**

I have a few questions:

* Please refer to the 'Weaknesses' section, and the concerns raised there.
* How does the 'complexity' of the dataset affect the relation between the Hessian eigenspace and the pruning map? For instance, suppose we consider models for classification - if the class-conditional distributions are poorly separated, does that decrease or increase the similarity between the pruning map and the Hessian eigenspace?
* Does the presence of complex interconnections (i.e. skip connections in ResNets) have an impact on the pruning mask, the Hessian eigenspace, and the similarity between the two?
* Have the authors tried to use the similarity between the Hessian eigenspace and the pruning mask to derive new pruning algorithms?

---

### Official Review · Reviewer_kBDH · 2023-10-31

**Soundness:** 1 poor
**Presentation:** 3 good
**Contribution:** 2 fair
**Rating:** 3
**Confidence:** 4

**Summary:**

This paper seeks to unify two lines of work in empirical deep learning.  The first is the emergence of traininable subnetworks or lottery tickets early in training, and the second is the gradient lying within the subspace of the top eigenvalues of the Hessian after a few initial training steps.  Since a pruning mask defines an axis-aligned subspace (i.e. a subspace whose dimensions align with the coordinate axes of the weights), the paper proposes comparing these two subspaces at different iterations in training.  The claimed conclusion is that there is high-overlap between these two subspaces after they both stabilize early in training.

**Strengths:**

The paper thoroughly considers which metric to use for comparing the pruning mask and top Hessian subspace. The distances proposed are not new but rather are various distances considered for the Grassmannian or manifold of $k$-dimensional linear subspaces.  Additional consideration is given to the fact that pruning masks occupy a subset of Grassmannian with the additional restriction that the subsapce be axis-aligned. Each metric is studied for randomly drawn matrices under two setups: (1) the overall dimension $D$ is fixed and the ratio of the subspace dimension $k$ to $D$ is varied and (2) the ratio $k/D$ is held constant and $D$ is increased.

The experiments with random matrices show that these metrics appear to converge as $D$ increases and can be sub-divided into "shrinking" metrics that go to 0 with increasing $D$ and "proportional" metrics which converge to a non-zero value.  In the setting of these experiments, the value the "proportional" metric converges to is determined by the ratio $k/D$.

Based on these experiments, the authors argue for using the overlap metric to compare pruning masks and top $k$ Hessian subspaces and use the value these metrics converge to with increasing $D$ as a baseline for how close we should expect random subspaces to be.  Overall, I thought the steps taken in this first part of the paper were reasonable.

**Weaknesses:**

(a) My primary concern about the paper is that while its claims about comparing subspaces are primarily empirical, results are only only shown for a small MLP (7030 parameters) on subsampled 16 x 16 MNIST.  The lottery ticket literature in particular shows that there are significant changes in behavior for large scale problems. [1], for example, shows that for larger problems (e.g. ResNet-50 on ImageNet) lottery tickets have to be found after a small amount of dense training rather than at random initialization.  Thus, my view is that supporting the paper's claims requires larger-scale experiments.

Note that I understand that the justification for the set up used is that computing the Hessian eigenspectra for larger networks is expensive. However, there are now a number of software tools aimed at studying the Hessian in large-scale networks; see for example PyHessian [2].  Also, I think just studying the overlap in the early parts of training would be sufficient.

(b) Next, I think the paper requires some clarification about the pruning mask that is being used in the experiment.  My understanding is at each step, the mask under consideration is some percentage of the largest magnitude weights in the current model. ($\rho$ is defined to be the ratio of unpruned parameters so does $\rho =0.2$ mean 80% of the weights are pruned?) This means that fundamentally the experiments compare how close the subspace of the $k$ highest magnitude weights is to the $k$ sharpest directions of the Hessian at each training iteration. Is this correct?

To make the comparison to the lottery ticket literature, the paper first needs a definition of how to determine whether a mask is a lottery ticket or not.  A standard definition would be that a mask is a lottery ticket if the associated subnetwork at initialization (or a few steps into dense training) is trainable to the same accuracy as the dense network.  Iterative Magnitude Pruning (IMP) typically uses a masks constructed at the end of training, so I think it is important to confirm the masks under consideration meet this definition by including the accuracy achieved when retraining the sparse network (or whatever metric would confirm this is a lottery ticket in your definition).  Note this could be done for a subset of the training steps; the general trend over training is what is important. *If these masks do not achieve a reasonable accuracy when retrained then these experiments do not tell us much about lottery tickets.*


(c) I found the applications the authors discussed beyond understanding the two phenomena to be unclear.  For example:

```
Since pruning masks can be obtained in linear time, our results suggest new ways for fast and effective low-rank Hessian approximations, with application to e.g. pruning and optimization methods as proposed by Hassibi et al.
```

Could the authors expand on this, i.e. give basic pseudocode for their idea?  Second, how could these results lead to "novel pruning algorithms" as described in the conclusion?

Minor Notes:
* Bottom of page 3: Reference to Pearlmutter (1994) should probably use citep rather than citet.
* Bottom of page 4: $\mathbb{B}^{D \times k}$ is never explicitly defined.  In addition to being binary, I think you need the criteria that there is only one non-zero per column.

References:

[1] Jonathan Frankle, Gintare Karolina Dziugaite, Daniel Roy, Michael Carbin.  "Linear Mode Connectivity and the Lottery Ticket Hypothesis." https://proceedings.mlr.press/v119/frankle20a

[2] Zhewei Yao, Amir Gholami, Kurt Keutzer, Michael Mahoney. "PyHessian: Neural Networks Through the Lens of the Hessian." https://arxiv.org/abs/1912.07145

**Questions:**

* In the definition of the chordal norm distance, should this say minimizing over orthogonal matrices $Z_i, Z_j$?  $A_1, A_2$ do not appear in the quantity you are minimizing.

* Do these results hold for any further iterations of IMP, i.e. what if you prune a further 20% of weights based on training the sparse subnetwork?

---

### Author Response · Authors · 2023-11-22
**Minor updates and clarifications**

We are grateful to the reviewers for their constructive and thoughtful feedback.
While the outlined strengths encourage us to continue with this line of work, we understand that substantial progress still needs to be done to address major concerns about the empirical nature of this work.
Scalability seems to be the main bottleneck, and we do not have a solution at this point (nor do libraries like PyHessian, as discussed in 2.1), but we are working on it, taking into consideration the suggested approaches.

Still, we wanted to take the chance to update the draft, to fix typos and clarify/expand on a few points that we feel can and should be addressed at this point (other points are duly noted and left for future work):

-------
### Authors' note: We did not perform pruning in this work, and we claim it is not desirable to establish the existence of this overlap

Reviewer kBDH asked: "Do these results hold for any further iterations of IMP, i.e. what if you prune a further 20% of weights based on training the sparse subnetwork?"

This seems to imply the misunderstanding that pruning was part of our methodology. We emphasize that in this work we did not perform any pruning, nor it was our original intention to do so. This was briefly noted in section 5 but we updated the draft to make this point clearer.

If we pruned the model during training, we would be restricting the parameter space, artificially forcing the overlap between masks and Hessian to be higher.
The fact that this overlap is high, even when parameters are not artificially confined to a subspace during training, reinforces the interest of our results.
We updated section 5 with a clarification.

-----
### In order to be talking about pruning masks, we need to show that the masks we are considering do lead to competitive performance

Reviewer kBDH mentioned: "If these masks do not achieve a reasonable accuracy when retrained then these experiments do not tell us much about lottery tickets".
Reviewer e5Zh mentioned: "The authors do not take into account the effect of different pruning criteria on the overlap similarity".

Despite our remark that we are not doing any pruning, this is a fair point: It remains a valid question to ask (albeit tangential to our results), whether crystallized masks are useful at all, and how useful they are depending on how they are obtained.

We plan to address this question with experiments in future work.

Furthermore, reviewer e5Zh further questioned our claim that the overlap between the masks and hessian eigenspaces suggests that large weights correspond to large curvatures in the loss.
This claim stems from the definition on the Grassmannians: obtaining large overlap measurements between top-Hessian eigenspace and magnitude masks implies directly that the largest parameters are majorly responsible for the largest directions of loss curvature.
But a very valid question is whether this is *specific* to parameter magnitudes, i.e. whether any other masking criteria would have the same effect. This is in fact not possible: since the columns of the eigenbasis are orthonormal, and overlap is a Frobenius norm of a subset of rows, high overlap with one mask criterium must necessarily mean low overlap with complementary criteria.

---------
### At high sparsities, the overlap is not significant (Fig. 4 has lower overlap than the random baseline for pruning ratio < 0.2)
This is unfortunately a misinterpretation of the results, due to lack of clarity in the figures.
Each different sparsity level has a different random baseline. For example, overlap has a random baseline of exactly k/D.
Which means that the baseline for rho=10% is 0.1, and not 0.2.

------------------
### what are the implications of the dependence of the measure's variance on the modality?  What are the implications of the expectation of overlap not depending on the modality?

As discussed, boolean random matrices introduce more variance in the low-$k$ regime.
This is indeed a valid concern when the overlap for high-sparsity masks is being studied, as relying only on the expectation may not be sufficient.
It may be observed in the O-vs-B plots for fixed $D$ and moving $r$ that this variance is fairly contained for proportional metrics, but this is indeed not clear for the smallest sparsities.
At the moment we do not have a definitive answer for this. We consider theoretical analysis of the variance, and running multiple seeds for future work.

---------------
### Is the overlap providing more insight than say dist_{p,F} or dist_{c,F}?

The advantage of overlap is that there is a closed-form expression for the baseline value, which equals exactly rho. We did not find such an expression for the other metrics, and thus we must rely on the baseline values from our synthetic experiments, gathered in Table 1.

---

### Meta-Review · Area_Chair_rSVV · 2023-12-14

**Metareview:**

The work investigates a possible connection between the top eigen-space of the Hessian and the active (axis parallel) parameter space of deep networks. While the reviewers feel that such a connection may be interesting, limited empirical evidence is provided to support the claim. Considerable additional work is needed to establish such a connection.

**Justification For Why Not Higher Score:**

The concerns shared by the reviewers are major and the authors seem to acknowledge the limitations.

**Justification For Why Not Lower Score:**

n/a

---

### Decision · Program_Chairs · 2024-01-16

Reject